# Atopic Dermatitis and Sensitisation to Molecular Components of *Alternaria*, *Cladosporium*, *Penicillium*, *Aspergillus*, and *Malassezia*—Results of Allergy Explorer ALEX 2

**DOI:** 10.3390/jof7030183

**Published:** 2021-03-04

**Authors:** Jarmila Celakovska, Radka Vankova, Josef Bukac, Eva Cermakova, Ctirad Andrys, Jan Krejsek

**Affiliations:** 1Department of Dermatology and Venereology, Faculty Hospital, Medical Faculty of Charles University, 50002 Hradec Králové, Czech Republic; 2Department of Clinical Immunology and Allergy, Faculty Hospital, Medical Faculty of Charles University, 50002 Hradec Králové, Czech Republic; vankovr@lfhk.cuni.cz (R.V.); ctirad.andrys@fnhk.cz (C.A.); jan.krejsek@fnhk.cz (J.K.); 3Department of Medical Biophysic, Medical Faculty of Charles University, 50002 Hradec Králové, Czech Republic; bukacjosef@seznam.cz (J.B.); cermakovae@lfhk.cuni.cz (E.C.)

**Keywords:** atopic dermatitis, molecular components, ALEX 2–Allergy Explorer, *Alternaria*, *Cladosporium*, *Penicillium*, *Aspergillus*, *Malassezia*, bronchial asthma, allergic rhinitis

## Abstract

Progress in laboratory diagnostics of IgE-mediated allergies is being made through the use of component-resolved diagnosis. The aim of our study is to analyze the sensitization profile to allergen reagents in patients suffering from atopic dermatitis with the use of the ALEX 2–Allergy Explorer and especially to show the sensitization to molecular components of molds and yeast. The complete dermatological and allergological examination including the examination of the sensitization to allergen reagents with Allergy Explorer ALEX 2 testing was performed. The relation between the sensitization to molecular components of molds and yeast and the severity of atopic dermatitis, and the occurrence of bronchial asthma and allergic rhinitis was evaluated. Altogether, 100 atopic dermatitis patients were examined—48 men and 52 women, with an average age of 40.9 years. The sensitization to Mala s 6, Mala s 11, Sac c, Asp f 6, Cla h and Cla h 8 correlates to the severity of atopic dermatitis. The sensitization to Sac c, Alt a 6, Cla h, Cla h 8 was observed significantly more frequently in patients suffering from bronchial asthma to Mala s 6 in patients suffering from allergic rhinitis. In patients with severe form of atopic dermatitis (AD), a very high level of specific IgE was recorded to Mala s 11 (in 36%) and to Asp f 6 (in 12%).

## 1. Introduction

Atopic dermatitis (AD), together with bronchial asthma and allergic rhinitis, belongs to so-called atopic diseases. Atopic dermatitis is a chronic severe itchy inflammatory disease of the skin. Genetic predisposition, abnormal immune response and defective skin barrier are involved in the etiopatogenesis of the disease. Interest in the disease is constantly increasing, due to its increasing prevalence and adverse effects on quality of life [1,2,3,4,5]. The prevalence in Western countries is 10–20% for children and 1–3% for adults [1,2]. Atopic dermatitis was originally considered to be mainly a childhood disease with an imbalance between the Th2 response and the escalated IgE response to external allergens. Today, atopic dermatitis is rated as a long-term disease with varying clinical manifestations and expressivity, with epidermal barrier disorder playing a central role [1,2,3,4,5]. A disorder of the skin barrier implies hydration, reparation and non-specific inflammation alertness; there is a higher susceptibility to bacterial colonization and viral infections. Transepidermal water loss (TEWL) and temperature are skin barrier function parameters that can be objectively measured and could help clinicians to evaluate disease severity accurately [1]. In atopic dermatitis patients, the skin barrier disorder arises as a result of a genetic defect for filagrin, involucrine and lorikrin [2,3,4,5]. This genetic defect also increases the risk of bronchial asthma in AD patients and also increases the risk of allergic rhinitis in both AD and non-AD patients. The abnormal expression of epidermal proteins caused by Th2-type protein cytokine allergens may increase the risk of sensibilization to these allergens and thus contribute to the development of atopic dermatitis [2,3,4,5]. Animal studies have shown that a defect in the skin barrier can lead to systemic sensibilization to allergens. Thymic stromal lymphopoietin (TSLP), IL-33, and IL-25 can control the transition from AD to asthma and food allergy [2,3,4,5].

Fungi (*Alternaria*, *Cladosporium*, *Penicillium* and *Aspergillus* species) are an important source of allergen molecules, involving various molecular structures including enzymes, toxins, cell wall components and highly preserved cross-reactive proteins. Fungi are very common in the environment and exposure to inhaled fungal allergens is almost constant throughout the year. Fungi can also colonize the human body and damage the airways by producing toxins, proteases, enzymes and other organic ingredients. The spectrum of allergic symptoms to fungi are allergic rhinitis, asthma and atopic dermatitis. Although numerous species of fungi have been associated with allergic diseases, the significance of fungi from the genera *Alternaria*, *Cladosporium*, *Penicillium*, *Aspergillus*, and *Malassezia* has been well documented [3]. *Alternaria* and *Cladosporium* species are considered to be important outdoor allergens, and sensitization and exposure to species of these genera is related to the development of asthma and rhinitis; xerophilic species of *Penicillium* and *Aspergillus*, excluding *Aspergillus fumigatus*, are indoor allergens [3]. *Malassezia* species are evaluated as the most common fungi found on human skin [4,5,6]. Other allergens, which play an important role in atopic dermatitis patients, are Der p 11 (*Dermatophagoides pteronyssinus*) and enterotoxin B from Staphylococcus aureus [4,5,6]. Despite its importance in the management of allergic diseases, the precise recognition of species-specific IgE sensitization to fungal allergens is often challenging because the majority of fungal extracts exhibit broad cross-reactivity with taxonomically unrelated fungi. Compared to other common environmental allergenic sources, such as pollens and dust mites, fungi are reported to be neglected and underestimated. Recent progress in gene technology has contributed to the identification of specific and cross-reactive allergen components from different fungal sources. However, data demonstrating the clinical relevance of IgE reactivity to these allergen components are still insufficient. Molecular allergy diagnosis using singleplex allergens or multiplex allergen microarrays are considered as the typical methods of precision medicine [6,7,8]. One of the multiplex metods is the ALEX 2–Allergy Explorer (ALEX^®^; MacroArray Diagnostics, Wien, Austria). The major advantage of ALEX 2 is the comprehensive IgE pattern obtained with a minute amount of serum [9,10]. With this multiplex allergy test, we receive the results of sensitization to 295 allergen reagents (117 allergenic extracts and 178 molecular components), so we can obtain the complete picture of the sensitization of each patient [9,10].

According to our previous studies with ISAC multiplex testing in AD patients, the majority of these patients have positive results of specific IgE to Phl p 1 (61%), to Bet v 1 (57%). Molecular components from PR-10 proteins, the NPC2 proteins family, Uteroglobin and Lipocalin, Alrernaria and Aspegillus play also the important role [11,12].

Aim of our study is to analyze the sensitization to allergens reagent (molecular components and allergen extracts) in atopic dermatitis patients and especially to show the sensitization to molecular components of molds and yeast (Mala s 5, Mala s 6, Mala s 11, Sac c, Alt a 1, Alt a 6, Asp f 1, Asp f 3, Asp f 4, Asp f 6, Cla h 8 and Pen ch) and to determine whether there are some differences between the sensitization profiles in mild, moderate and severe forms of AD and in subgroups of patients suffering from bronchial asthma and allergic rhinitis.

## 2. Material and Methods

### 2.1. Patients and Methods

In the period from 2018–2020, 100 patients suffering from atopic dermatitis at the age of 14 years and older were examined. All these patients were examined in the Department of Dermatology, Faculty Hospital Hradec Králové, Charles University, Hradec Králové, Czech Republic. The diagnosis of atopic dermatitis was made with the Hanifin–Rajka criteria [13]. The inclusion criteria was: atopic dermatitis as defined by the criteria of Hanifin and Rajka [13]. Systemic treatment with antihistamines and topical treatment were allowed. The exclusion criteria was: systemic therapy (cyclosporin, systemic corticoids, biological therapy), pregnancy, and breastfeeding. Patients with atopic dermatitis with other systemic diseases were excluded from the study as well. A complete dermatological and allergological examination was performed on patients included in the study.

This study was approved by Ethics committee of the Faculty Hospital Hradec Králové, Charles University of Prague, Czech Republic.

### 2.2. Dermatological Examination

A complete dermatological examination was performed on patients included in the study. This examination was performed by dermatologists in the Department of Dermatology, Faculty Hospital Hradec Králové, Charles University, Czech Republic. The severity of atopic dermatitis was scored in agreement with SCORAD, with the assessment of topography items (affected skin area), intensity criteria and subjective parameters [14]. To measure the extent of atopic dermatitis, the rule of nines was applied on a front/back drawing of the patient’s inflammatory lesions. The extent was graded from 0–100 points. The intensity part of the SCORAD index consists of six items: erythema, oedema/papules, excoriations, lichenification, crusts, and dryness. Each item was graded on a scale of 0–3. The subjective items included daily pruritus and sleeplessness. Both subjective items were graded on a 10-cm visual analog scale and the maximum subjective score was 20 points. All items were filled out in the SCORAD evaluation form. The SCORAD index formula was: A/5 + 7B/2 + C. In this formula, A is defined as the extent (0–100 points), B is defined as the intensity (0–18 points), and C is defined as the subjective symptoms (0–20 points). The severity of atopic dermatitis is evaluated with SCORAD as a mild form up to 20 points, as moderate over 20 to 50 points, and as a severe form over 50 points [14]. The evaluation of SCORAD score was performed every two months during the study [14].

### 2.3. Examination of Specific IgE to Molecular Components

We measured the serum level of the specific IgE (sIgE) by the Multiplex test ALEX 2–Allergy Explorer (ALEX^®^; MacroArray Diagnostics, Wien, Austria [10,15,16]. The measuring range for the specific IgE was 0.3–50 kU_A_/L (quantitative) and for total IgE was 1–2500 kU/L (semiquantitative). The results are expressed as Class 0 (<0.3 kU_A_/L-negative), Class 1 (0.3–1 kU_A_/L—low level), Class 2 (1–5 kU_A_/L—moderate level), Class 3 (5–15 kU_A_/L—high level), and Class 4 (>15 kU_A_/L—very high level) [10].

Initially, allergens were coupled to activated nanoparticles, for coupling individual and combinatorial optimization. Each allergen was attached reflecting its biochemical properties and specific requirements for stability, thereby preserving the full epitope complexity. The nanoparticles multiplied the surface of the solid-phase presenting the allergen during the immunoassay, enabling highly sensitive detection. In the next step, the allergen-bearing nanoparticles were deposited onto a solid-phase matrix, forming a macroscopic array of individual assay parameters. The different allergens and components, spotted onto a nitrocellulose membrane as immunosorbent in a cartridge chip, were incubated with 0.5 mL of a 1:5 dilution of serum under agitation, the serum diluent containing a cross-reactive carbohydrate determinants (CCDs) inhibitor. After incubation for 2 h, the chips were extensively washed. A pre-titered dilution of anti-human IgE labeled with alkaline phosphatase was added and incubated for 30 min. Following another washing cycle, the enzyme substrate was added, and after a few minutes, the reaction was complete. After the membranes were dried, the quantification of this colorimetric enzyme assay was performed with an easy-to-use and affordable image explorer. The image acquisition and analysis of a single test took only a few seconds. The assay time was 3.5 h, and tests per run are up to 50 per operator, with manual processing [9,10].

### 2.4. Allergological Examination

#### 2.4.1. Bronchial Asthma

The diagnosis of bronchial asthma (AB) was determined according to the guidelines of the Global Initiative for Asthma (GINA) at allergy outpatients clinic of the Institute of Clinical Immunology and Allergology, Faculty Hospital Hradec Kralove, Czech Republic (Global Initiative for Asthma. Global Strategy for asthma management and prevention—Updated 2015. www.ginasthma.com accessed on: 15 February 2021).

#### 2.4.2. Allergic Rhinitis

The evaluation of allergic rhinitis (AR) was made according to the allergy testing and personal history of the Institute of Clinical Immunology and Allergology, Faculty Hospital Hradec Kralove, Czech Republic [17]. AR was defined as a process which included 3 cardinal symptoms during one last year: sneezing, nasal obstruction, and mucus discharge. The symptoms occur with allergen exposure in the allergic patient [17].

### 2.5. Statistical Analysis

The analysis of the results of specific IgE in the examination of ALEX 2–Allergy Explorer testing was performed. We tested whether there were significant differences in the prevalence of sensitization to molecular components of molds and yeast in mild, moderate, and severe forms of AD, in patients suffering from bronchial asthma, and allergic rhinitis. In the case of severity of AD, we formed a 2 by 3 tables (negative, positive level of specific IgE in relation to mild, moderate and severe form of AD) and 5 by 3 tables (the classes of specific IgE 0, 1, 2, 3, 4 in relation to mild, moderate and severe form of AD). In the case of bronchial asthma and allergic rhinitis, we formed 2 by 2 tables. Such a table was evaluated by the Fisher’s exact test. The significance level was set to 5%. The *p*-value is displayed in the right-hand column. Unfortunately, zeroes appeared in many of the tables and one has to be careful in making conclusions.

## 3. Results

We show the characteristics of patients in Table 1. Our study included 100 patients—52 women and 48 men, with the average age of 40.9 years, and the average SCORAD 39, s.d.13.1 points.

We recorded the mild form of AD in 14 patients (14%), the moderate form in 61 patients (61%), and the severe form in 25 patients (25%). The diagnosis of bronchial asthma was made in 55 patients (55%); the diagnosis of allergic rhinitis was made in 74 patients (74%).

Altogether, 295 allergen reagents (117 allergenic extracts and 178 molecular components) were examined in these patients. The order of allergen reagents (allergenic extracts and molecular components) in 100 atopic dermatitis patients according to the frequency is recorded in Table 2. The highest sensitization rate to grass-species specific component Phl p 1 (Timothy, beta-expansin) was recorded in 57% of patients. We observed high sensitization to other molecular components and extracts in 49% of patients to Fag s 1 (European beech), in 48% to Cor a 1.0103, Cyn d (*Bermuda grass*, Beta-expansin), Cor a pollen (Hazel pollen), in 45% to Der f 2 (House dust mite, NPC2 family), Fra a 1+3 (PR 10 protein, strawberry), Phl p 2 (grass group II, Timothy) in 44% to Cor a 1.0401 (Hazelnut, PR 10 protein), Der p 2 (House dust mite, NPC2 family), Fel d 1 (Cat, Uteroglobin), in 43% to Aln g 1 (Alder, PR 10 protein), in 42% to Pas n (*Bahia grass*), Phl p 5.0101, Phl p 6 (Timothy), and in 41% to Lep d (Storage mite, NPC2 family 2), Mal d 1 (Apple, PR 10 protein), Table 2.

We show the results of sensitization to Mala s 5, Mala s 6, Mala s 11, Sac c, Alt a 1, Alt a 6, Asp f 1, Asp f 3, Asp f 4, Asp f 6, Cla h 8 and Pen ch (Table 3). The highest sensitization rate was observed to Alt a 1 (*Alternaria*) in 26% of patients and to Mala s 11 (Manganese superoxide dismutase, *Malassezia sympodialis*) in 24% of patients. The sensitization to Asp f 6 (*Aspergilus fumigatus*, Mn superoxide dismutase) was recorded in 20% of patients; to Asp f 3 (*Aspergilus fumigatus*, *Peroxysomal protein*) and to Cla h 8 (Mannitol dehydrogenase, *Cladosporium herbarum*) in 15%; and to Sac c (*Saccharomyces cerevisiae*) and to Mala s 6 (*Malassezia sympodialis*, Cyclophilin) in 14%. The sensitization to Alt a 6 (Enolase, *Alternaria alternata*) was recorded in 12% and to Mala s 5 in 10% of patients. A low sensitization rate was observed to Asp f 1, Asp f 4 and to Pen ch (*Penicillium chrysogenum*) in 3–4% of patients. In patients with a severe form of AD, the occurrence of positive specific IgE to molecular components Mala s 6, Mala s 11 and Sac c from yeast, Asp f 6, Cla h and Cla h 8 from molds was recorded significantly more frequently (*p* < 0.05), **(**Table 3).

We made the statistical analysis of the relation between the severity of AD and the level of specific IgE (classes 0, 1, 2, 3, 4) to molds and yeast. The significant relation was confirmed between the severity of atopic dermatitis and the level of specific IgE to these molecular components and allergen extract: Mala s 11 (*p*-value = 0.00342), Asp f 3 (*p*-value = 0.00479), Asp f 4 (*p*-value = 0.04229), Cla h *(p*-value = 0.01598) and Cla h 8 (*p*-value *=* 0.02364). Although we have not confirmed the significant relation between the severity of AD and the level of specific IgE (classes 0–4) to Mala s 6, Asp f 6 and to allergen extract Sacc according to the level of specific IgE (classes 0, 1, 2, 3, 4), this difference being nearly significant (*p*-value = 0.064091 for Mala s 6, *p*-value =0.05590 for Sacc and *p*-value = 0.05773 for Asp f 6), Table 4.

In the subgroup of patients suffering from bronchial asthma, the positive level of specific IgE to molecular components Sac c, Alt a 6, Cla h, Cla h 8 were observed significantly more frequently, (*p <* 0.05), (Table 5).

In the subgroup of patients suffering from allergic rhinitis, the positive specific IgE to molecular component Mala s 6 was observed significantly more frequently, (*p <* 0.05), (Table 5).

## 4. Discussion

We compared our results with other studies from the Middle-European region with the outcomes describing the sensitization patterns to molecular components in patients suffering from atopic disease [12,18,19,20]. Our results are in agreement with the hypothesis [18,19,20,21], that grasses (Phl p 1) and *Betulaceae* (Bet v 1) components comprise the vast majority of pollen sensitizations in the condition of the Middle-European region. Regarding the sensitization to molds and yeast in our study, the highest sensitization rate was observed to Alt a 1 (*Alternaria*) in 26% of patients and to Mala s 11 (Manganese superoxide dismutase, *Malassezia sympodialis*) in 24% of patients. We confirmed the significantly higher sensitization to Mala s 6, Mala s 11, Sac c, Asp f 6, Cla h and Cla h 8 in patients suffering from a moderate and severe form of AD. Our results confirm the important role of Mala s 11 in patients suffering from a moderate and severe form of AD; in these patients, a very high level of specific IgE (class 4) to Mala s 11 was recorded in 36% of patients and a moderate level of specific IgE (class 2) was recorded in 16% of patients. The important role of patients suffering from severe form of AD play also the molecular components of *Aspergillus fumigatus*; in these patients, a high level of specific IgE (class 3) to Asp f 6 was recorded in 24% of patients, a very high level of specific IgE (class 4) to Asp f 6 was recorded in 12% of patients, a moderate level of sIgE (class 2) to Asp f 3 was recorded in 20% of patients. We also recorded the significance of *Cladosporium herbarum*; in patients with a severe form of AD, a low level of specific IgE (class 1) to Cla h was recorded in 16% and Cla h 8 in 28.0% of patients.

The sensitization to molecular components such as Sac c, Alt a 6, Cla h, Cla h 8 was observed with the significantly higher occurrence in the subgroup of patients with bronchial asthma. So, our results show that sensitization to Sac c, Cla h and Cla h 8 was recorded with the significantly higher occurrence both in patients suffering from a severe form of AD and from bronchial asthma. Similarly, the simultaneous sensitization to Mala s 6 was recorded with the significantly higher occurrence both in patients suffering from a severe form of AD and allergic rhinitis. There is the possibility that direct contact of skin with these allergens could trigger signals to initiate a Th2 allergic response and that the epithelial cell-derived cytokines such as TSLP, IL-33, and IL-25 may drive the progression from atopic dermatitis to bronchial asthma [21]. In 2014, it was reported that IgE antibodies to Der p 11 are more common in sera from patients with atopic dermatitis [2,21].

Compared to the literature, our results show a lower number of patients with positive levels of sIgE to molecular components of *Malassezia allergens*. According to some studies, the sensitization rate to Mala s 5 is unknown, the sensitization to Mala s 6 is in 92% of AD patients and to Mala s 11 in 43–75% of patients [8,22,23,24,25,26,27]. *Malassezia* spp. can affect the course of atopic dermatitis by several pathogenic mechanisms, such as releasing more allergens in lower pH (pH < 6); this environment is typical for AD. Another mechanism is the cross-reactivity, because there is similarity in fungal thioredoxin and human proteins. *Malassezia* spp. and keratinocytes are in interaction, which may alter the release of cytokines. In addition, there is sequence homology between Asp f 6 and Mala s 11, so the specific IgE against Asp f 6 can be evaluated as a marker for autoreactivity [28].

Brodska [25] et al. investigated the relationship between AD and sensitization to *Malassezia antigens*. They assessed 173 patients with AD. The total serum IgE and specific IgE to *Malassezia* were determined and correlated with the clinical picture of AD, sex, age, and the EASI [25].

The sensitization to *Saccharomyces cerevisiae* (Sac c) was recorded in our study in 14% of patients; the sensitization was significantly higher in patients with a severe form of AD and in the subgroup of patients suffering from bronchial asthma. Regarding the classes of specific IgE to Sac c, a low level was recorded in 16% of patients suffering from a severe form, and a moderate level of specific IgE was also recorded in 16.0% of patients suffering from a severe form. On the other hand, Kortekangas-Savolainen et al. observed a positive skin prick test reaction in 94% of patients with severe AD, in 76% with moderate AD, and in 25% with mild AD or no history of AD [29].

In our study, the sensitization to Penicillium Pen ch was low without some differences according to the severity of AD and according to the occurrence of bronchial asthma and allergic rhinitis.

The prevalence of sensitization to fungi in young atopic patients in relation to age and clinical importance was studied [30]. The prevalence of specific IgE for *Cladosporium* ranked first, followed closely by *Aspergillus* and *Alternaria.* The question is why sensitization to *Alternaria alternata* is a risk factor for asthma and also, why the severity of asthma is associated to this mold. The importance and relative contribution of fungal sensitization to airway disease, compared with the other allergens, remains to be established [30]. Recent research on the identification and characterization of *Alternaria alternata* allergens has allowed for the consideration of new perspectives in the categorization of allergenic molds, assessment of exposure and diagnosis of fungi-induced allergies [31]. In our study, we also confirmed that sensitization to Alt a 6 is significantly higher in subgroup of patients suffering from bronchial asthma. Although we have not confirmed the significant difference in the sensitization to molecular component Alt a 6 in patients suffering from a moderate and severe form of AD, this difference is nearly significant (*p*-value *=* 0.07285). Regarding the level of specific IgE to Alt a 6, we recorded in 12% of patients suffering from a severe form of AD a high level of specific IgE.

In our previous study [32], we evaluated the sensitization to fungi in AD patients and we also evaluated the relation of fungal sensitization to the occurrence of other atopic diseases and parameters. The sensitization to fungi was recorded in 100 patients (30%). According to our results, the occurrence of bronchial asthma, rhinitis, family history of atopy, sensitization to grass and trees was significantly higher in AD patients with a sensitization to fungi. No relation was found between the severity of AD and the sensitization to fungi [32]. It should be pointed out that in this previous study, we evaluated the serum level of specific IgE to the extract mixture of fungi [32].

The disrupted epidermal barrier can allow the penetration of allergens of molds and yeast. Subsequently, a T helper 2 inflammatory response develops; thymic stromal lymphopoietin significantly acts here as the mediator in the spreading of inflammation at distant sites, including the airways. The results of our study, as well from other studies, show that the proper function of the epidermal barrier is absolutely essential in patients with atopic dermatitis.

## 5. Conclusions

In patients suffering from AD, the high sensitization was recorded to Alt a 1 (*Alternaria*) in 26% and to Mala s 11 (Manganese superoxide dismutase, *Malassezia sympodialis*) in 24% of patients. The sensitization to Mala s 6, Mala s 11, Sac c, Asp f 6, Cla h and Cla h 8 correlates to the severity of atopic dermatitis. The sensitization to Sac c, Alt a 6, Cla h, Cla h 8 was observed significantly more frequently in patients suffering from bronchial asthma to Mala s 6 in patients suffering from allergic rhinitis. In patients with a severe form of AD, a very high level of specific IgE was recorded to Mala s 11 (in 36%) and to Asp f 6 (in 12%).

The disruption in the skin epidermal barrier permits allergen sensitization and colonization by pathogens and may facilitate the atopic march.

## Figures and Tables

**Table 1 jof-07-00183-t001:** The characteristic of patients with atopic dermatitis.

Table 100	
Patients suffering from atopic dermatitis	100 patients
Sex	48 men, 52 women
The average age	40.9 years (min. age 14 years, max. age 67 years)
The average SCORAD	39 points, s.d. 13.1 points
Severity of AD	mild form 14 (14%)
	moderate form 61 (61%)
	severe form 25 (25%)
Asthma bronchiale	55 (55%)
Allergic rhinitis	74 (74%)

**Table 2 jof-07-00183-t002:** The order of allergen reagents (allergenic extracts and molecular components) according to the frequency in 100 atopic dermatitis patients (it is shown the sensitization in the frequency 10% and more). The sensitization to molecular components of molds and yeast (Mala s 5, Mala s 6, Mala s 11, Sac c, Alt a 1, Alt a 6, Asp f 1, Asp f 3, Asp f 4, Asp f 6, Cla h 8 and Pen ch) is recorded extra bold *. Low sensitization rate was observed to Asp f 1, Asp f 4 and to Pen ch (*Penicillium chrysogenum*) in 3–4% of patients.

Allergen Reagents (Allergenic Extracts and Molecular Components)	Frequency in %
Phl p 1 (Beta-expansin, *Timothy grass*)	57.0
Bet v 1 (PR-10 protein, birch), Lol p 1 (Beta-expansin, Rye grass), Sec c_pollen (cultivated rye, pollen)	53.0
Fag s1 (PR-10 protein, European beech)	49.0
Cor a 1.0103 (PR-10 protein, hazel pollen) Cyn d 1 (Beta-expansin, *Bermuda grass*)	48.0
Cor a_pollen (hazel pollen)	47.0
Der f 2 (NPC2 family, house dust mite) Fra a 1+3 (PR 10 protein + Non-specific lipid transfer protein, type 1, strawberry) Phl p 2 (Expansin, *Timothy grass*)	45.0
Cor a 1.0401 (PR-10 protein, hazel pollen) Cyn d (*Bermuda grass*) Der p 2 (NPC2 family, house dust mite) Fel d 1 (Uteroglobin, cat)	44.0
Aln g 1 (PR-10 protein, alder)	43.0
Pas n (*Bahia grass*) Phl p 5.0101 (Grass Group 5/6, *Timothy grass*), Phl p 6 (Grass Group 5/6, *Timothy grass*)	42.0
Lep d 2 (NPC2 family, storage mite) Mal d 1 (PR-10 protein, apple)	41.0
Can f 1 (Lipocalin, dog) Der p 23 (Peritrophin-like protein domain, house dust mite)	36.0
Gly m 4 (PR-10 protein, soybean)	35.0
Der p 1 (Cysteine protease, house dust mite)	34.0
Ara h 8 (PR-10 protein, peanut) Gly d 2 (NPC2 family, storage mite)	33.0
Art v (Mugwort), Can f 6 (Lipocalin, dog)	32.0
Der f 1 (Cysteine protease, house dust mite) Fel d 7 (Lipocalin, cat)	31.0
Can f_male urine (male dog urine)	30.0
Can f 4 (Lipocalin, dog) Equ c 1 (Lipocalin, horse)	29.0
Pla l (English plantain)	28.0
Ory c 3 (Uteroglobin, rabbit) Phr c (Common reed)	27.0
Aca s (storage mite) **Alt a 1 (unknown, *Alternaria alternata*) *** Der p 5 (unknown, house dust mite) Fel d 4 (Lipocalin, cat)	26.0
Amb a (ragweed) Tyr p (storage mite)	25.0
Ach d (house cricket) Der p 7 (Mites group 7, Bactericidal permeability-increasing-like protein, house dust mite) Loc m (*Migratory locust*) **Mala s 11 (Mn superoxide dismutase, *Malassezia sympodialis*) ***	24.0
Amb a 4 (Plant defensin, ragweed) Der p 20 (Arginine kinase, house dust mite), Ten m (Mealworm)	23.0
Api g 1 (PR-10 protein, celery), Mus m 1 (Lipocalin and urinary prealbumin, house mouse)	22.0
Bla g 9 (Arginine kinase, German cockroach) Can f_Fd1 (Uteroglobin, dog), Cav p 1 (Lipocalin, Guinea pig)	21.0
**Asp f 6 (Mn superoxide dismutase, *Aspergillus fumigatus*) *** Bla g 4 (Calycin, Lipocalin, German cockroach) Fra e 1 (Ole e 1-like protein family, European ash), Jug r_pollen (walnut pollen)	20.0
Art v 1 (Plant defensin, Mugwort) Dau c (Carrot) Dau c 1 (PR-10 protein, carrot)	19.0
Par j 2 (Non-specific lipid transfer protein, type 1, Pellitory of the wall) Pen m 2 (Arginine kinase, shrimp) Ves v 5 (Antigen 5, Yellow jacket venom)	18.0
Ama r (redroot pigweed) Can f 2 (Lipocalin, dog) Rat n (rat)	17.0
Api m (Honey bee venom) Cup a 1 (Pectate Lyase, cypress) Fra e (European ash), Gal d_white (egg white) Per a (American cockroach)	16.0
Aca m (Acacia) Api m 1 (Phospholipase A2, Honey bee venom) **Asp f 3 (*Peroxysomal protein*, *Aspergillus fumigatus*) *** **Cla h 8 (*Mannitol dehydrogenase*, *Cladosporium herbarum*) *** Cry j 1 (Pectate lyase, Japanese cedar) Der p 21 (unknown, house dust mite) Hel a (sunflower seed) Hom g (lobster) Pla l 1 (Ole e 1-like protein family, English plantain) Sal k (Russian thistle, saltwort)	15.0
Act d 2 (Thaumatin-like protein, kiwi fruit) **Mala s 6 (Cyclophilin, *Malassezia sympodialis*) *** **Sac c (*Saccharomyces cerevisiae*) *** Sol t (potato)	14.0
Api m 10 (Icarapin Variant 2, Honey bee venom) Pha v (Green bean, French bean)	13.0
**Alt a 6 (Enolase, *Alternaria alternata*) *** Amb a 1 (Pectate lyase, ragweed), Api g 2 (Non-specific lipid-transfer protein, type 1, celery) Blo t 5 (Mites group 5, storage mite) Can s (hemp), Hor v (barley) *Lol* spp. (squid) Pan b (Northern shrimp), Phl p 7 (Polcalcin, *Timothy grass*) Phod s 1 (Lipocalin, Siberian hamster) Sal k 1 (Pectin methylesterase, Russian thistle, saltwort) Urt d (nettle)	12.0
Bet v 2 (Profilin, birch) Cuc m 2 (Profilin, muskmelon) *Pec* spp. (Scallop) Sec c_flour (cultivated rye) Ses i 1 (2S albumin, sesame) Tyr p 2 (NPC2 family, storage mite) Ves v (Yellow jacket venom), Ves v 1 (Phospholipase A1, Yellow jacket venom) Zea m 14 (Non-specific lipid transfer protein, type 1, maize)	11.0
Che q (quinoa) **Mala s 5 (unknown, *Malassezia sympodialis*) *** Ole e 1 (Ole e 1-family, olive) Pers a (avocado) Phl p 12 (Profilin, *Timothy grass*) Pla a 2 (Polygalacturonase, London plane tree) Pol d (paper wasp venom) Pyr c (pear) Sola l (tomato) Ulm c (Elm) Zea m (Maize)	10.0

**Table 3 jof-07-00183-t003:** The sensitization rate to molecular components of yeast and molds in 100 atopic dermatitis patients. We show the number of positive results of specific IgE in the whole group of patients and in patients suffering from mild, moderate and severe form of AD (relative frequency of sensitization rate is recorded in %). The significance level was set to 5%. The significant difference (*p*-value *<0.05*) is shown extra bold *, (the statistical analysis is made with Fisher´s exact test).

	Number of Patients (%) with Positive Results of Specific IgE in the Whole Study—100 Patients (=100%)	Severity of Atopic Dermatitis			
Yeast		Mild Form 14 Patients (=100 %)	Moderate Form 61 Patients (=100 %)	Severe Form 25 Patients (=100 %)	*p*-Value
**Mala s 5** *Malassezia sympodialis*	10 (10.0%)	0	5(8.2%)	5 (20.0%)	0.123
**Mala s 6** *Malassezia sympodialis*, Cyclophilin,	14 (14.0%)	0	6 (9.8%)	8 (32.0%)	**0.011 ***
**Mala s 11** Manganese superoxide dismutase, *Malassezia sympodialis*	24 (24.0%)	0	11 (18.0%)	13 (52.0%)	**0.001 ***
**Sac c** *Saccharomyces cerevisiae*	14 (14.0%)	0 (0%)	6 (9.8%)	8 (32.0%)	**0.011 ***
**Moulds**					
**Alt a 1** ***Alternaria***	26 (26.0%)	3 (21.4%)	16 (26.2%)	7 (28.0%)	0.949
**Alt a 6** Enolase, *Alternaria alternata*	12 (12.0%)	0	6 (9.8%)	6 (24.0%)	0.073
**Asp f 1** Aspergilus fumigatus Mitogillin family	4 (4.0%)	0	1(1.6%)	3 (12%)	0.098
**Asp f 3** Aspergilus fumigatus *Peroxysomal protein*	15 (15.0%)	0	8 (13.1%)	7 (28.0%)	0.055
**Asp f 4** Aspergilus fumigatus	3 (3.0%)	1 (1.7%)	0	2 (8.0%)	0.057
**Asp f 6** Aspergilus fumigatus Mn superoxide dismutase	20 (20.0%)	0	10(16.4%)	10 (40.0%)	**0.007 ***
**Cla h** *Cladosporium herbarum*	6 (6.0%)	0	1 (1.6%)	5 (20.0%)	**0.008 ***
**Cla h 8** Mannitol dehydrogenase, *Cladosporium herbarum*	15 (15.0%)	0	7(11.5%)	8(32.0%)	**0.017 ***
**Pen ch** *Penicillium chrysogenum*	3 (3.0%)	0	1(1.6%)	2 (8.0%)	0.204

**Table 4 jof-07-00183-t004:** The number of patients according to the classes of specific IgE to molecular components and allergen extracts of yeast and molds and according to the severity of AD (mild, moderate, severe form). The relative frequency of sensitization in the classes of specific IgE 0–4 in patients suffering from mild, moderate and severe form of AD is recorded. The significance level was set to 5%. The significant difference (*p*-value < 0.05) is shown extra bold *, the statistical analysis is made with Fisher´s exact test. The measuring range for specific IgE is 0.3–50 kU_A_/L (quantitative) and for total IgE is 1–2500 kU/L (semiquantitative). The results are expressed as Class 0 (<0.3 kU_A_/L—negative), Class 1 (0.3–1 kU_A_/L—low level), Class 2 (1–5 kU_A_/L—moderate level), Class 3 (5–15 kU_A_/L—high level), and Class 4 (>15 kU_A_/L—very high level) [10].

		Number of Patients (Relative Frequency in %)			
	Classes of Specific IgE	Mild Form 14 Patients (=100%)	Moderate Form 61 Patients (=100%)	Severe Form 25 Patients (=100%)	*p-*Value
**Mala s 5** *Malassezia sympodialis*	0	14 (100%)	56 (91.8%)	20 (80.00%)	0.435
	**1**	0	1 (1.64%)	0	
	**2**	0	2 (3.28%)	1 (4.00%)	
	**3**	0	0	1 (4.00%)	
	**4**	0	2 (3.28%)	3 (12.00%)	
**Mala s 6** *Malassezia sympodialis*, Cyclophilin,	0	14 (100%)	55 (90.16%)	17 (68.0%)	0.065
	**1**	0	1 (1.64%)	4 (16.00%)	
	**2**	0	3 (4.92%)	3 (12.00%)	
	**3**	0	2 (3.28%)	1 (4.00%)	
	**4**	0	0	0	
**Mala s 11** Manganese superoxide dismutase, *Malassezia sympodialis*	0	14 (100%)	50 (81.97%)	12 (48%)	
	**1**	0	0	0	**0.003 ***
	**2**	0	3 (4.92%)	4 (16.00%)	
	**3**	0	1 (1.64%)	0	
	**4**	0	7 (11.48%)	9 (36.00%)	
**Sac c** *Saccharomyces cerevisiae*	0	14 (100%)	55 (90.16%)	17 (68.00%)	
	**1**	0	3 (4.92%)	4 (16.00%)	0.056
	**2**	0	3 (4.92%)	4 (16.00%)	
	**3**	0	0	0	
	**4**	0	0	0	
**Alt a 1** ***Alternaria***	0	11 (78.57%)	45 (73.77%)	18 (72.00%)	0.182
	**1**	0	0	0	
	**2**	0	0	0	
	**3**	2 (14.29%)	1 (1.64%)	1 (4.00%)	
	**4**	1 (7.14%)	15 (24.59%)	6 (24.00%)	
**Alt a 6** Enolase, *Alternaria alternata*	0	14 (100%)	55 (90.16%)	19 (76.00%)	
	**1**	0	1 (1.64%)	1 (4.00%)	0.393
	**2**	0	3 (4.92%)	1 (4.00%)	
	**3**	0	1 (1.64%)	3 (12.00%)	
	**4**	0	1 (1.64%)	1 (4.00%)	
**Asp f 1** Aspergilus fumigatus Mitogillin family	0	14 (100%)	60 (98.36%)	22 (88.00%)	
	**1**	0	1 (1.64%)	1 (4.00%)	0.233
	**2**	0	0	1 (4.00%)	
	**3**	0	0	1 (4.00%)	
	**4**	0	0	0	
**Asp f 3** *Aspergilus fumigatus* *Peroxysomal protein*	0	14 (100%)	53 (86.89%)	18 (72.00%)	
	**1**	0	7 (11.48%)	1 (4.00%)	**0.005 ***
	**2**	0	0	5 (20.00%)	
	**3**	0	1 (1.64%)	0	
	**4**	0	0	1 (4.00%)	
**Asp f 4** Aspergilus fumigatus	0	13 (92.86%)	61 (100%)	23 (92.0%)	
	**1**	0	0	0	
	**2**	1 (7.14%)	0	0	**0.042 ***
	**3**	0	0	1 (4.00%)	
	**4**	0	0	1 (4.00%)	
**Asp f 6** *Aspergilus fumigatus* Mn superoxide dismutase	0	14 (100%)	51 (83.61%)	15 (60.0%)	
	**1**	0	2 (3.28%)	0	
	**2**	0	3 (4.92%)	1 (4.00%)	0.058
	**3**	0	3 (4.92%)	6 (24.00%)	
	**4**	0	2 (3.28%)	3 (12.00%)	
**Cla h** *Cladosporium herbarum*	0	14 (100%)	60 (98.36%)	20 (80.00%)	
	**1**	0	1 (1.64%)	4 (16.00%)	**0.016 ***
	**2**	0	0	1 (4.00%)	
	**3**	0	0	0	
	**4**	0	0	0	
**Cla h 8** Mannitol dehydrogenase, *Cladosporium herbarum*	0	14 (100%)	54 (88.52%)	17 (68.00%)	
	**1**	0	7 (11.48%)	7 (28.00%)	**0.024 ***
	**2**	0	0	1 (4.00%)	
	**3**	0	0	0	
	**4**	0	0	0	
**Pen ch** *Penicillium chrysogenum*	0	14 (100%)	60 (98.36%)	23 (92.00%)	0.204
	**1**	0	1 (1.64%)	2 (8.00%)	
	**2**	0	0	0	
	**3**	0	0	0	
	**4**	0	0	0	

**Table 5 jof-07-00183-t005:** The sensitization rate to molecular components of yeast and molds in 100 atopic dermatitis patients. We show the number of positive results in the whole group of patients and in patients with bronchial asthma (AB) and allergic rhinitis (AR), the relative frequency of sensitization rate is recorded in %. The significance level was set to 5%. The significant difference (*p*-value < 0.05) in the sensitization rate is shown extra bold *, (the statistical analysis is made with Fisher´s exact test).

		Bronchial Asthma (AB)		*p-*Value	Allergic Rhinitis (AR)		*p-*Value
	Number of Patients (%) with Positive Results of Specific IgE in the Whole Study—100 Patients (=100%)	AB Yes 55 Patients (=100%)	AB No 45 Patients (=100%)		AR Yes 76 Patients (=100%)	AR No 24 Patients (=100%)	
**Mala s 5** *Malassezia sympodialis*	10 (10.0%)	5 (9.1%)	5 (11.1%)	0.750	7 (9.5%)	3 (11.5%)	0.718
**Mala s 6** *Malassezia sympodialis*, Cyclophilin,	14 (14.0%)	6 (10.9%)	8 (17.8%)	0.391	14 (18.9%)	0	**0.018 ***
**Mala s 11** Manganese superoxide dismutase, *Malassezia sympodialis*	24 (24.0%)	13 (23.6%)	11 (24.4%)	1.000	21 (28.4%)	3 (11.5%)	0.111
**Sac c** *Saccharomyces cerevisiae*	14 (14.0%)	12 (21.8%)	2 (4.4%)	**0.018 ***	11 (14.9%)	3 (11.5%)	1.000
**Moulds**							
**Alt a 1** ***Alternaria***	26 (26.0%)	16 (29.1%)	10 (22.2%)	0.497	22 (29.7%)	4(15.4%)	0.198
**Alt a 6** Enolase, *Alternaria alternata*	12 (12.0%)	11 (20.0%)	1(2.2%)	**0.011 ***	10 (13.5%)	2 (7.7%)	0.726
**Asp f 1** *Aspergilus fumigatus* Mitogillin family	4 (4.0%)	3 (5.5%)	1 (2.2%)	0.625	4 (5.4%)	0	0.571
**Asp f 3** Aspergilus fumigatus *Peroxysomal protein*	15 (15.0%)	10 (18.2%)	5 (11.1%)	0.405	13 (17.6%)	2 (7.7%)	0.342
**Asp f 4** *Aspergilus fumigatus*	3 (3.0%)	1 (1.8%)	2 (4.4%)	0.587	2 (2.7%)	1 (3.8%)	1.0000
**Asp f 6** *Aspergilus fumigatus* Mn superoxide dismutase	20 (20.0%)	13(23.6%)	7 (15.6%)	0.452	18 (24.3%)	2 (7.7%)	0.089
**Cla h** *Cladosporium herbarum*	6 (6.0%)	6 (10.9%)	0	**0.031 ***	6 (8.1%)	0	0.335
**Cla h 8** Mannitol dehydrogenase, *Cladosporium herbarum*	15 (15.0%)	12 (21.8%)	3 (6.7%)	**0.045 ***	14 (18.9%)	1 (3.8%)	0.107
**Pen ch** *Penicillium chrysogenum*	3 (3.0%)	3 (5.5%)	0	0.250	3 (4.1%)	0	0.566

## Data Availability

Not Availability.

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
