# Peer review of "Atopic Dermatitis and Sensitisation to Molecular Components of Alternaria, Cladosporium, Penicillium, Aspergillus, and Malassezia—Results of Allergy Explorer ALEX 2"

_jof, 2021, doi:10.3390/jof7030183_

Round 1

Reviewer 1 Report

English should be improved.

Abstract

Line 12 “to analyze” is written twice

The abstract should be a single paragraph and should follow the style of structured abstracts but without headings.

Introduction

The introduction should be shorten.

Line 30 . Recent research regarding defective skin barrier in atopic dermatitis should be added. Skin Barrier Function in Psoriasis and Atopic Dermatitis: Transepidermal Water Loss and Temperature as Useful Tools to Assess Disease Severity. J Clin Med. 2021 Jan 19;10(2):359. doi: 10.3390/jcm10020359. PMID: 33477944; PMCID: PMC7833436.

Line 87. It is said that the relation between allergic diseases and the fungi investigated in this article is well documented, so what this study adds?

Line 94. …the an important role. Atopic dermatitis……

Line 95 an important role

Line 107. Could you explain how ALEX 2 works?

Aim of the study should be included as a final paragraph in the introduction. Statistical analysis, patients and methods and dermatological examination should be included in Material and Methods

Line 127. Instead of occurrence it would be better to say prevalence

Line 133. If more than 20% of cells have expected frequencies < 5, the Fisher’s exact test would be applied

Line 138. Was the atopic dermatitis diagnosis made by a dermatologist?

Line 139. Only atopic dermatitis patients receiving topic treatments were allowed to enter in the study? Why were patients receiving systemic therapies excluded?

Results

Table 1 should be showed in a table format, not in lines

Line 162-165. The SOCORAD is too high if only AD patients with topical treatments were allowed to enter in the study. Too many patients with moderate and severe AD with only topical treatments. Why?

Table 3. The p value should be included in other column and not in a line

Discussion

The first paragraph should sump up the most important results

Line 220. There is no need to repeat the objectives. The discussion is to sum up the result and discuss them with previous reports. “The purpose….. allergic rhinitis” (line 220-224) should be eliminated.

Line 229-234. It should be expressed in a more understandable way

References numbers in the text should be placed in square brackets.

Reference list should be follow the same style (if possible following ACS style guide)

Author Response

Dear reviewer,

 thank you very much for the evaluation of this article. I appreciate your comments very much and I do my best to improve the manuscript according to your recommendations.

line 12 “to analyze” is written twice

– it is corrected

The abstract should be a single paragraph and should follow the style of structured abstracts but without headings

- it is corrected

Introduction - The introduction should be shorten.

- it is corrected, the introduction is shorten

Line 30 . Recent research regarding defective skin barrier in atopic dermatitis should be added. Skin Barrier Function in Psoriasis and Atopic Dermatitis: Transepidermal Water Loss and Temperature as Useful Tools to Assess Disease Severity. J Clin Med. 2021 Jan 19;10(2):359. doi: 10.3390/jcm10020359. PMID: 33477944; PMCID: PMC7833436.

- this citation is completed

Line 87. It is said that the relation between allergic diseases and the fungi investigated in this article is well documented, so what this study adds?

We examined in detail the importance of molecular components of molds and yeast in atopic march. There are no studies dealing with the question in patients suffering from atopic dermatitis with the use of multiplex examination ALEX2 Allergy expolerer.

We confirmed that the occurrence of positive results of specific IgE to molecular components Mala s 6, Mala s 11, Asp f 6, Cla h 8 and to allergen extracts Sacc and Cla h (without the regards to the classes of specific IgE) correlates significantly with the severity of AD.

We performed also the statistical analysis of the relation between the severity of AD and the level of specfic IgE of molecular components of molds and yeast according to the level of specific IgE (classes 0, 1, 2, 3, 4). The significant relation between the severity of atopic dermatitis and the level of specific IgE (classes 0, 1, 2, 3, 4) to these molecular components was confirmed: Mala s 11, Asp f 3,  Asp f 4, Cla h 8 and allergen extract Cla h.  

Our results confirmes the important role of Mala s 11 in patients suffering from moderate and severe form of AD; in these patients, the very high level of specific IgE (class 4) to Mala s 11 was recorded in 36 % of patients and moderate level of specific IgE (class 2) was recorded in 16 % of patients. Important role in patients suffering from severe form of AD play also the molecular components of Aspergillus fumigatus. In these patients, the  high level of specific IgE (class 3) to Asp f 6  was recorded in 24 % of patients, very high level of specific IgE (class 4) to Asp f 6 was recorded in 12 % of patients, the moderate level of sIgE (class 2) to Asp f 3 was recorded in 20 % of patients. We recorded also the significance of Cladosporium; in patients with severe form of AD, the low level of specific IgE (class 1) to Cla h was recorded in 16 % and Cla h 8 in 28.0 %  of patients.

The low level of specific IgE (class 1) and moderate level (class 2) to Sacc was recorded in 16.0 %  of patients suffering from severe form – the difference is nearly significant.

I have completed the discussion with the results of our previous study:

In our previous study [32]  we evaluated  the sensitization to fungi in AD patients and we also evaluated the relation of fungal sensitization to the occurrence of other atopic diseases and parameters. The sensitization to fungi was recorded in 100 patients (30%). According to our results, the occurrence of asthma bronchiale, rhinitis, family history about atopy, sensitization to grass and trees was significantly higher in AD patients with sensitization to fungi. No relation was found between the severity of AD and the sensitization to fungi [32]. It should be pointed out that in this previous study we evaluated the serum level of specific IgE to mixture of fungi.

 Line 94. …the an important role. Atopic dermatitis

- it is corrected

Line 95 an important role - it is corrected

Line 107. Could you explain how ALEX 2 works? – it is explained

Aim of the study should be included as a final paragraph in the introduction. Statistical analysis, patients and methods and dermatological examination should be included in Material and Methods

- it is corrected according to your recommendations

Line 127. Instead of occurrence it would be better to say prevalence

- it is corrected

Line 133. If more than 20% of cells have expected frequencies < 5, the Fisher’s exact test would be applied.

Our results were now calculated according to the Fisher’s exact test. We show the Table 3 a) with the calculation with Fisher exact test. The p values are a little different from previous calculations, but this test shows the same significance as the previous one, so the Fisher ´s exact test confirmes our previous calculations. According to the recommendation of Reviewer´s No.2, we analysed if the classes of specific IgE (0, 1, 2, 3, 4) correlates to the severity of AD. 

 Line 138. Was the atopic dermatitis diagnosis made by a dermatologist?

- yes, it is completed in the text

Line 139. Only atopic dermatitis patients receiving topic treatments were allowed to enter in the study? Why were patients receiving systemic therapies excluded?

- the inclusion citeria are completed. The severity of AD is significantly influenced by systemic therapy (cyclosporin, systemic corticoids, biological therapy).

Results

Table 1 should be showed in a table format, not in lines

- it is corretcted to Table format

Line 162-165. The SOCORAD is too high if only AD patients with topical treatments were allowed to enter in the study. Too many patients with moderate and severe AD with only topical treatments. Why?

-  we included especially patients with moderate and severe form to be examined with the ALEX 2  multiplex testing to find the triggering factors for the exacerbations of AD.The severity of AD is significantly influenced by systemic therapy (cyclosporin, systemic corticoids, biological therapy). After evaluating the results of ALEX 2 examination, we recommended the specific regimen (diet regimen, elimination of allergens, imonotherapy...). The systemic treatment was recommende later to patients with refractory forms of AD according to the guidelines.

Table 3. The p value should be included in other column and not in a line

- it is corrected. The Table 3 is noe divided in 3 tables, p values are recorded in the column.

Discussion

The first paragraph should sump up the most important results -- it is corrected

 Line 220. There is no need to repeat the objectives. The discussion is to sum up the result and discuss them with previous reports. “The purpose….. allergic rhinitis” (line 220-224) should be eliminated.

-- it is corrected

 Line 229-234. It should be expressed in a more understandable way

References numbers in the text should be placed in square brackets.

- it is corrected

Reference list should be follow the same style (if possible following ACS style guide)

Reviewer 2 Report

The authors analysed the prevalence of sensitisation to molecular components of several fungi in atopic dermatitis. I have several comments regarding the study:

  1. Please provide the cut offs values for severity in SCORAD.
  2. Bronchial asthma and allergic rhinitis were diagnosed in some subjects. Does it mean that the patients were symptomatic? If not, was it based on the past medical history? If yes, when symptoms were observed for the last time.
  3. Please include also the classes of specific IgE levels - it would provide more information regarding the dependence of AD severity and particular sensitivity to allergens.
  4. In the discussion please focuse only on fungi allergens as it was the aim of the study. Do not conclude with following statement: "Atopic dermatitis patients suffer mainly from sensitisation to these molecular components: Phl p 1 (Timothy, beta-expansin) (...)" as it was not the aim of the study. 
  5. Please change "asthma bronchiale" to "bronchial asthma".

Author Response

Review 2

Dear reviewer,

 thank you very much for the evaluation of this article. I appreciate your comments and I do my best to improve the manuscript according to your recommendations.

The authors analysed the prevalence of sensitisation to molecular components of several fungi in atopic dermatitis. I have several comments regarding the study:

  1. Please provide the cut offs values for severity in SCORAD.

         - it is corrected

  1. Bronchial asthma and allergic rhinitis were diagnosed in some subjects. Does it mean that the patients were symptomatic? If not, was it based on the past medical history? If yes, when symptoms were observed for the last time. We completed the section of methods:    Allergological examination

Bronchial asthma The diagnosis of bronchial asthma (AB), was determined according to the guidelines of the Global Initiative for Asthma (GINA) at allergy outpatients clinic of the Institute of Clinical Immunology and Allergology, Faculty Hospital Hradec Kralove, Czech Republic (Global Initiative for Asthma. Global Strategy for asthma management and prevention – Update 2015. www.ginasthma.com).

Allergic rhinitis

The evaluation of allergic rhinitis (AR), was made according to the allergy testing and personal history of the Institute of Clinical Immunology and Allergology, Faculty Hospital Hradec Kralove, Czech Republic (17). AR was defined as a process which included 3 cardinal symptoms during last one year: sneezing, nasal obstruction, and mucus discharge. Symptoms occur with allergen exposure in the allergic patient (17).

  1. Please include also the classes of specific IgE levels - it would provide more information regarding the dependence of AD severity and particular sensitivity to allergens. – we included new tables with the classes of specific IgE levels (0, 1, 2, 3, 4) to provide more information regarding the dependence of AD severity

 Table 3b)

We made the statistical analysis of the relation between the severity of AD and the level of specific IgE (classes 0, 1, 2, 3, 4) to molds and yeast.  The significant relation was confirmed between the severity of atopic dermatitis and the level of specific IgE to these molecular components and allergen extract: Mala s 11 (p – value = 0.00342), Asp f 3  (p – value = 0.00479), Asp f 4 (p – value = 0.04229), Cla h (p – value = 0.01598) and Cla h 8 (p – value = 0.02364).

Although we have not confirmed the significant relation between the severity of AD and the level of specific IgE (classes 0-4) to Mala s 6 , Asp f 6  and to allergen extract Sacc according to the level of specific IgE (classes 0, 1, 2, 3, 4),  this difference being nearly significant (p – value = 0.064091 for Mala s 6,  p – value =0.05590 for Sacc and p – value = 0.05773 for Asp f 6).

 Our results confirmes the important role of Mala s 11 in patients suffering from moderate and severe form of AD; in these patients, the very high level of specific IgE (class 4) to Mala s 11 was recorded in 36 % of patients and moderate level of specific IgE (class 2) was recorded in 16 % of patients. Important role in patients suffering from severe form of AD play also the molecular components of Aspergillus fumigatus. In these patients, the  high level of specific IgE (class 3) to Asp f 6  was recorded in 24 % of patients, very high level of specific IgE (class 4) to Asp f 6 was recorded in 12 % of patients, the moderate level of sIgE (class 2) to Asp f 3 was recorded in 20 % of patients. We recorded also the significance of Cladosporium; in patients with severe form of AD, the low level of specific IgE (class 1) to Cla h was recorded in 16 % and Cla h 8 in 28.0 %  of patients.

The low level of specific IgE (class 1) and moderate level (class 2) to Sacc was recorded in 16.0 %  of patients suffering from severe form – the difference is nearly significant.

  1. In the discussion please focuse only on fungi allergens as it was the aim of the study. Do not conclude with following statement: "Atopic dermatitis patients suffer mainly from sensitisation to these molecular components: Phl p 1 (Timothy, beta-expansin) (...)" as it was not the aim of the study.

it is corrected    

  1. Please change "asthma bronchiale" to "bronchial asthma". – It is corrected

Round 2

Reviewer 1 Report

Table 1.  This information should be divided (number of patients, sex, age, SCORAD). You can use different rows (Age: 40.9 years; SCORAD 39) or you can mention the in the text and do not repeat them in a table

Only two or three decimals should be used for p values

English correction is needed

Author Response

Review 1

Dear reviewer,

 thank you very much for the evaluation of this article once more. I appreciate your comments very much and I do my best to improve the manuscript according to your recommendations.

Table 1.  This information should be divided (number of patients, sex, age, SCORAD). You can use different rows (Age: 40.9 years; SCORAD 39) or you can mention the in the text and do not repeat them in a table.

The different rows are used in the Table 1.

Table 1. The characteristic of patients  with atopic dermatitis.

                                                            The characteristic of patients, number of patients 

Patients suffering from atopic dermatitis

100 patients  

Sex

48 men, 52  women

The average age

40.9 years (min. age 14 years,  max. age 67 years)

The average SCORAD

39 points,  s.d. 13.1 points

Severity of AD

mild form  14 (14 %)

moderate form  61 (61 %)

severe form 25 (25 %)

Asthma bronchiale

55 (55 %)

Allergic rhinitis 

74 (74 %)

Only two or three decimals should be used for p values

it is corrected

Once more, thank you very much for the review.

Author

Reviewer 2 Report

None

Author Response

Dear reviewer,

 thank you very much for the evaluation of this article once more. I appreciate it very much.